# The Autophagy Process in Cervical Carcinogenesis: Role of Non-Coding-RNAs, Molecular Mechanisms, and Therapeutic Targets

**DOI:** 10.3390/cells11081323

**Published:** 2022-04-13

**Authors:** Alfredo Lagunas-Martínez, Vicente Madrid-Marina, Claudia Gómez-Cerón, Jessica Deas, Oscar Peralta-Zaragoza

**Affiliations:** 1Direction of Chronic Infections and Cancer, Research Center in Infection Diseases, National Institute of Public Health, Av. Universidad No. 655, Cerrada los Pinos y Caminera, Colonia Santa María Ahuacatitlán, Cuernavaca 62100, Morelos, Mexico; alagunas@insp.mx (A.L.-M.); vmarina@insp.mx (V.M.-M.); jessicadeas@gmail.com (J.D.); 2Research Center in Population Health, Department of Cancer Epidemiology, National Institute of Public Health, Av. Universidad No. 655, Cerrada los Pinos y Caminera, Colonia Santa María Ahuacatitlán, Cuernavaca 62100, Morelos, Mexico; cceron@insp.mx

**Keywords:** autophagy, cervical cancer, HPV E5, E6, E7, ncRNAs, signaling pathways

## Abstract

Autophagy is a highly conserved multistep lysosomal degradation process in which cellular components are localized to autophagosomes, which subsequently fuse with lysosomes to degrade the sequestered contents. Autophagy serves to maintain cellular homeostasis. There is a close relationship between autophagy and tumor progression, which provides opportunities for the development of anticancer therapeutics that target the autophagy pathway. In this review, we analyze the effects of human papillomavirus (HPV) E5, E6, and E7 oncoproteins on autophagy processes in cervical cancer development. Inhibition of the expression or the activity of E5, E6, and E7 can induce autophagy in cells expressing HPV oncogenes. Thus, E5, E6, and E7 oncoproteins target autophagy during HPV-associated carcinogenesis. Furthermore, noncoding RNA (ncRNA) expression profiling in cervical cancer has allowed the identification of autophagy-related ncRNAs associated with HPV. Autophagy-related genes are essential drivers of autophagy and are regulated by ncRNAs. We review the existing evidence regarding the role of autophagy-related proteins, the function of HPV E5, E6, and E7 oncoproteins, and the effects of noncoding RNA on autophagy regulation in the setting of cervical carcinogenesis. By characterizing the mechanisms behind the dysregulation of these critical factors and their impact on host cell autophagy, we advance understanding of the relationship between autophagy and progression from HPV infection to cervical cancer, and highlight pathways that can be targeted in preventive and therapeutic strategies against cervical cancer.

## 1. Introduction

Autophagy is a highly conserved process in eukaryotic cells, which serves to degrade and recycle aging biological macromolecules and damaged organelles. It includes macroautophagy, the most prevalent form of autophagy, as well as micro-autophagy and chaperone-mediated autophagy, processes that differ with respect to their physiological functions and the mode of cargo delivery to the lysosome [1]. Eukaryotic organisms remove senescent, degenerative, and necrotic cells, as well as subcellular structures, through autophagy to maintain homeostasis. The autophagy process has an adaptive role in the protection of organisms against diverse pathologies, including viral infections and carcinogenesis. The molecular cascade that regulates and executes autophagy has been the subject of recent comprehensive reviews [2,3,4].

Autophagy involves the delivery of cytoplasmic cargo sequestered inside double-membrane vesicles to the lysosome for degradation. Initial steps include the formation (vesicle nucleation) and expansion (vesicle elongation) of an isolation membrane, also known as a phagophore. The edges of the phagophores then fuse (vesicle completion) to form the autophagosome, a double-membraned vesicle that sequesters the cytoplasmic material. This is followed by fusion of the autophagosome with a lysosome to form an autolysosome where the captured material, together with the inner membrane, is degraded (Figure 1) [1].

One of the key regulators of autophagy is the mammalian target of rapamycin (mTOR) kinase, which provides the major inhibitory signal to shut off autophagy in the presence of growth factors and abundant nutrients [5]. Downstream of the mTOR kinase, there are more than 20 genes identified in yeast, known as autophagy-related genes (ATGs), which encode proteins essential for the execution of autophagy. Many of these genes are evolutionarily conserved in mammalian cells [6].

The regulation of autophagy overlaps closely with signaling pathways that regulate tumorigenesis. Several tumor suppressor genes involved in upstream inhibition of mTOR signaling, including PTEN, TSC1, and TSC2, stimulate autophagy; conversely, mTOR-activated oncogene products such as class I PI3K and Akt inhibit autophagy. P53, the most commonly mutated tumor suppressor gene in human cancers, positively regulates autophagy in DNA-damaged cells through AMPK activation of the TSC1/TSC2 complex and subsequent inhibition of mTOR, or via upregulation of damage-regulated autophagy modulator (DRAM), a lysosomal protein that may induce autophagy [7,8]. Death-associated protein kinase (DAPK), a protein with tumor- and metastasis-suppressing properties, also induces autophagy and apoptosis, and is commonly silenced by methylation in human cancers [9]. The Bcl-2 and Bcl-XL oncoproteins, which are overexpressed in many cancers, are thought to mediate oncogenesis by suppressing mitochondrial membrane permeabilization, one of the rate-limiting steps of apoptosis. In addition, endoplasmic reticulum (ER)-localized Bcl-2 and Bcl-XL inhibit autophagy by binding to the Beclin-1 autophagy protein [10]. In some viral infections and associated pathologies, the expression of autophagy-related genes is regulated by ncRNAs. ncRNA expression profiling in cervical cancer has allowed the identification of autophagy-related ncRNAs whose expression is associated with HPV infection and cervical carcinogenesis. A growing body of evidence implicates ncRNAs in the regulation of autophagy in cervical cancer. We analyzed the function of ncRNAs in the regulation of autophagy, to characterize the relationship between autophagy and progression from HPV infection to cervical cancer.

In summary, there is a strong correlation between molecules involved in the induction of autophagy and tumor suppression, and between molecules involved in the inhibition of autophagy and oncogenesis. In this review, we discuss recent evidence that clarifies and supports the importance of autophagy, the role of HPV E5, E6, and E7 oncoproteins, and the function of ncRNAs in the regulation of autophagy, to characterize the relationship between autophagy and progression from HPV infection to cervical cancer. Understanding the role of autophagy in the development of cervical cancer is of great significance, and autophagy regulation represents a new potential therapeutic target in cervical cancer. 

## 2. Autophagy in Premalignant Cervical Lesions and Cervical Cancer

Cervical precancerous lesions and cervical cancer constitute a major public health problem worldwide. Molecular, epidemiological, and clinical investigations have identified human papillomavirus (HPV) as the main cause of cervical dysplasia and cervical cancer [11]. Despite the availability of vaccines which protect against infection with oncogenic HPV types, HPV-associated carcinogenesis is expected to remain a major health problem for decades to come, particularly in low- and middle-income countries with limited public health infrastructure [12]. HPVs are small DNA viruses that infect epithelial tissues, replicate in the stratified layers of skin and mucosa, and may give rise to benign lesions such as warts or condylomata acuminata, or lead to malignancy. HPVs are classified as either high-risk (hr-HPV) or low-risk (lr-HPV) types based on their clinical associations. The hr-HPV types, such as HPV16 and HPV18, are associated with lesions that can progress to high-grade intraepithelial neoplasia and ultimately to cancer. In contrast, the lr-HPV types, such as HPV6 and HPV11, are associated primarily with condylomata acuminata, also known as anogenital warts, which very rarely undergo malignant transformation [13].

There are two efficient host antiviral responses which often cooperate to limit viral propagation throughout the organism: inflammation and self-destruction by apoptosis and autophagy. Hr-HPVs can counteract inflammation and apoptosis to keep their host cells alive and maximize viral reproduction [14]. The hr-HPV genomes encode for at least three proteins with growth-stimulating and transforming properties: E5, E6, and E7. The E5 protein contributes to cellular transformation by increasing the mitogenic stimulus from growth factor receptors to the nucleus [15]. The E6 and E7 proteins stimulate cellular progression through the G1/S transition despite the presence of various G1 arrest signals in their host cells. The best-described target for E6 is the p53 tumor suppressor protein. Binding of E6 to p53 promotes p53 degradation through a ubiquitin-dependent mechanism [16]. The E7 protein is best known for its interaction with the pRb tumor suppressor protein [17]. The E2 gene encodes a regulatory protein that negatively regulates the viral promoter that directs the expression of the E6 and E7 oncogenes in genital HPVs [18]. A characteristic of hr-HPV-related carcinogenesis is the role of HPV-induced epigenetic changes, as well as an integration of the viral genome into the host chromosomes in a manner that results in the loss of viral E2 trans-repressor protein expression, thereby maintaining high levels of E6 and E7 expression [19]. These observations suggest that the loss of E2 expression may be an important step in HPV-associated carcinogenesis. It has also been reported that antiapoptotic caspase-8/FLICE inhibitory protein (c-FLIP) expression is increased in E2-disrupted cervical lesions compared with cervical lesions that express E2, suggesting that the overexpression of c-FLIP occurs mainly after HPV genome integration. Interestingly, MiR-126 reverses drug resistance to TRAIL by inhibiting the expression of c-FLIP [20]. The E6 and E7 oncoproteins are necessary for maintenance of the transformed cellular phenotype, but their expression alone is not sufficient to transform human cells. Additional alterations (multiple “hits”) are required for full malignant transformation of HPV-infected cells [21]. It is important to note that not all persistent hr-HPV infections lead to cervical carcinogenesis. One mechanism for the elimination of these hr-HPV-infected cells is apoptosis [22]. The role of HPV proteins in apoptosis has been widely documented [23].

We now shift focus to autophagy, an essential cellular mechanism which has a “housekeeping” role in normal physiological processes. These processes include removal of old, aggregated, and misfolded proteins, clearance of damaged organelles, growth regulation, and aging. Autophagy is involved in a variety of biological functions including development, cellular differentiation, defense against pathogens, response to nutritional starvation, and genome maintenance [24,25]. Autophagy in epithelial cells is a form of programmed cell death that involves cellular recycling via the degradation of cellular constituents in lysosomes. Autophagy also maintains homeostasis by providing cytoprotection, including cell remodeling, recovery, and renewal, through the repair of cell damage and removal of hazardous aggregates of proteins, whole organelles such as mitochondria, or intracellular pathogens. In tumors, some central areas have inadequate vascularization, leading to nutrient and oxygen deprivation and metabolic stress. Autophagy is triggered under these conditions, promoting the metabolism of cellular components to provide nutrients, thus ameliorating metabolic stress and allowing for cell survival and tumor growth [26]. In these areas of the tumor, mTOR is deactivated, inducing the autophagy process; conversely, reactivation of mTOR inhibits autophagy, so there exists an inverse relationship between mTOR signaling and autophagy [27]. Molecules produced by autophagy such as carbohydrates, lipids, and nucleotides can provide substrate for glycolysis, the tricarboxylic acid cycle (TCA) or the pyrophosphate pathway [28]. Therefore, autophagy has a significant role in propagating carbon metabolism. Cancer cells can rewire carbon metabolism to adapt to nutrient deprivation through the autophagy process.

Regarding the relationship between HPV infection and autophagy, several studies have described the effect of HPV infection on the regulation of autophagy-related genes (ATG genes). Even lr-HPV can regulate the expression of ATG genes in condylomata acuminata. Jiang et al. reported that two key ATG genes that encode proteins involved in the formation of autophagosomes were downregulated in condylomata acuminata, and there were no differences in expression among different HPV genotypes [29]. Recent research has provided new insights into the autophagic response to HPV infection. The HPV11 E6 protein activates autophagy by repressing the Akt/mTOR and Erk/mTOR circuits. In contrast to the hr-HPV E6 gene, the lr-HPV11 E6 gene does not affect the expression of p53 [30]. UBC9, the key enzyme of the small ubiquitin-like modifier (SUMO) pathway, was shown to be upregulated during pre-cancerous and cancerous stages of cervical transformation [31]. The physiological mechanism by which UBC9 is degraded in cells through autophagy is altered in the presence of HPV E6 and E7 oncoproteins, via inhibition of autophagosome–lysosome fusion [31].

In addition, autophagy is closely linked to apoptosis induced by ER stress. There are common upstream signaling pathways between autophagy and apoptosis induced by ER stress, including PERK/ATF4, IRE1 alpha, ATF6, and Ca^2+^. Autophagy can block the induction of apoptosis by inhibiting the activation of apoptosis-associated caspases, which could reduce cellular injury; however, it can also help to induce apoptosis. In addition, the activation of apoptosis-related proteins can inhibit autophagy by degrading autophagy-related proteins such as Beclin-1, ATG4D, ATG3, and ATG5. Although common upstream signaling pathways have been found while investigating the interactions of various autophagy- and apoptosis-related proteins, the regulatory mechanisms remain poorly understood. Autophagy clearly has a dual role, and there are interactions and shared regulatory mechanisms between autophagy and apoptosis under ER stress conditions [32]. Autophagy is not only a compensatory mechanism under physiological and stress conditions; it is also an important process in cancer development, either acting alone or through crosstalk with the apoptosis process.

## 3. The Role of HPV E5, E6, and E7 Oncoproteins in the Autophagy Process in Cervical Cancer 

Autophagy is a defense mechanism that protects the cell against intracellular pathogens. Nevertheless, viruses have developed strategies to inhibit both apoptosis and autophagy [33]. Some reports suggest that hr-HPVs inhibit autophagy to promote cervical carcinogenesis [34,35]. The mechanism is not clear; however, some proteins that play an important role in autophagy, such as Beclin-1 and microtubule-associated protein 1A/1B light chain 3B (LC3B), are downregulated in human cervical squamous cell carcinoma [36]. These findings suggest that the degradation of these autophagy markers favors hr-HPV infection. In support of this hypothesis, HPV16 infectivity has been shown to be inhibited by high levels of autophagy induced during the entry of HPV16 virions into human primary keratinocytes. HPV16 is able to successfully infect cells only after knockout of genes that have an important role in autophagy such as PIK3C3 and ATG7 [35]. Similarly, Surviladze et al. reported that after exposure of HPV16 pseudovirions to human keratinocytes, Akt and mTOR pathways were activated. The activation of these pathways inhibited autophagy and facilitated viral infection [37].

It is well known that the hr-HPV genotypes are necessary to maintain HPV infection; however, the genetic background of the host also plays an important role in this process. A recent case-control study found that several SNPs of the ATG4 gene were associated with the risk of cervical cancer. The interaction of the SNPs rs807181 and rs807183 appeared to confer a risk of HPV infection in this group of patients, suggesting that the ATG4A gene may be important in susceptibility to cervical cancer [38]. To understand how HPV inhibits autophagy, we describe in the following section the role of early viral oncoproteins in autophagy inhibition.

The HPV early viral proteins interact with a plethora of cellular proteins involved in different cellular functions. As mentioned previously, the main hr-HPV proteins that contribute to growth-stimulating and transforming properties are E5, E6, and E7. E5 contributes to cellular transformation by interfering with the pathways of several tyrosine kinase receptors. E6 and E7 promote immortalization and transformation by blocking the function of several cellular proteins, including the tumor suppressor proteins p53 and pRb [39,40]. Belleudi et al. reported that HPV16 E5 protein inhibits autophagy in human keratinocytes [41]. In this study, the investigators found that in E5 oncoprotein-expressing HaCaT cells, E5 regulates autophagy through a decrease in keratinocyte growth factor receptor (KGFR) levels. Furthermore, the E5 protein inhibits serum starvation-induced autophagy. The mechanism through which HPV16 E5 inhibits autophagy is not fully understood, but several key proteins have been identified. E5 was found to inhibit the degradation of the autophagy substrate SQSTM1/p62 and reduce the levels of LC3-II, an autophagosome marker. This same group determined that E5 interferes with the assembly mechanism of the autophagosome. Furthermore, it has been reported that E5 protein from bovine delta papillomavirus interacts with SQSTM1/p62 and LC3-II [42]. However, it is unknown at present whether this same interaction occurs with HPV16 E5 protein. 

Another important mechanism through which HPV16 E5 oncoprotein may inhibit autophagy is alteration of the expression of autophagy genes. Under basal conditions or after a pro-autophagic stimulus, HPV16 E5 has been reported to downregulate the expression of key autophagy genes (Becn1, ATG5, and LC3) and p53-target autophagy genes (ULK1, ATG4a, and ATG7) in HaCaT cells [41].

The expression of HPV E6 and E7 oncoproteins is necessary for cell transformation. In this section, we describe how E6 and E7 oncoproteins inhibit autophagy. It has been demonstrated that UBC9 (E2-conjugating enzyme required for SUMOylation) is upregulated in low-grade squamous intraepithelial lesions (L-SIL) and high-grade SIL (H-SIL). HPV16 E6 and E7 expression promotes UBC9 upregulation and accumulation of SUMO1 (a member of the SUMO protein family) in human keratinocytes. E6 and E7 proteins from lr-HPVs (HPV6, HPV10, HPV11) also upregulate UBC9 expression [31]. The regulation of UBC9 expression by HPV16 E6 and E7 is p53-dependent. UBC9 silencing by siRNAs induces apoptosis in HPV16 E6 and E7-expressing keratinocytes, suggesting that UBC9 is necessary for apoptosis resistance in these cells. In addition, several experiments have shown that UBC9 is degraded by autophagy in different epithelial cells including gSAOS, MCF7, HaCaT, and human keratinocytes, suggesting that UBC9 is generally degraded through an autophagosomal process. HPV16 E6 and E7 oncoproteins reduce fusion between autophagic vacuoles and lysosomes in human keratinocytes. In premalignant cervical lesions, there is an increase in SQSTM1/p62 expression, indicating a decrease in autophagic activity [31].

Our group has demonstrated that restoration of p53 levels is mediated by MG132 and activation of the CD95 pathway through apoptosis antigen-1 (APO-1) antibody, which is responsible for the induction of autophagy in HPV16 E6-expressing keratinocytes [43]. Furthermore, we found that an increase in LC3 mRNA correlates with lipidation of autophagy-related protein LC3B and induction of autophagy. These data suggest that restoration of p53 levels (and its potential activation as a transcriptional factor) favors apoptosis and autophagy in HPV16 E6-expressing keratinocytes.

Chen et al. found a significant correlation between the expression of the protein anti-autophagy factor (ATAD3A) and the presence of hr-HPV in cervical cancer samples [44]. They demonstrated that knockdown of HPV E6 and E7 expression in cell lines derived from squamous cell carcinomas (SKG-I) decreased ATAD3A protein levels. Furthermore, silencing of ATAD3A by siRNAs activated autophagy, reflected in an increase in the number of autophagosomes. Interestingly, it has been suggested that ATAD3A participates in drug resistance in cancer cells [45]. This evidence supports the theory that HPV oncoproteins inhibit autophagy and promote drug resistance through an increase in ATAD3A protein levels.

Hanning et al. reported that after silencing of HPV16, E6, and E7 by siRNAs, an increase in autophagy gene expression (SASP, TPP1, STX12, TAP1, and others) was observed in keratinocytes (W12 cells) [46]. Consistent with the expression of autophagy-related genes, HPV E6 and E7 gene silencing resulted in autophagy induction characterized by conversion of LC3B-I to the lipidated form LC3B-II and reduction in SQSTM1/p62 protein levels (Figure 2). In contrast, HPV16 E6 and E7 genes have been reported to activate autophagy through ATG9B and LAMP1 genes in SiHa and CaSki cells [47]. These conflicting findings could be explained by the different cell types as well as the phase of transformation.

It appears that the inhibition of autophagy by E6 and E7 oncoproteins by HPV16 is not an exclusive feature of the alphapapillomavirus genus; other genera such as the betapapillomavirus genus can produce similar effects. Akgül et al. demonstrated that co-expression of E6 and E7 proteins from HPV8 (a papillomavirus associated with squamous cell carcinoma of the skin) downregulates checkpoint kinase-1 (CHK1) protein levels in keratinocytes. A decrease in CHK1 and LC3B protein levels was observed in E6/E7-expressing organotypic skin cultures, suggesting inhibition of autophagosome maturation due to downregulation of LC3B [48]. In contrast to HPV16 and HPV8 E6 and E7 proteins, HPV11 E6 activates autophagy in keratinocytes, decreasing phosphorylation levels of mTOR, Akt, and Erk [28]. In summary, the evidence suggests that each HPV oncogene has a unique effect on the modulation of the autophagy process in cervical cancer cells.

It has been reported that hr-HPV proteins have substantially different functions than lr-HPV proteins. For instance, hr-HPV E5 activates the epidermal growth factor receptor (EGFR), induces activation of PI3K, and downregulates MHC-class I [17]. E6 from hr-HPVs binds to p53 and promotes its degradation through a ubiquitin-dependent mechanism [49] and activates hTERT [50]. HPV produces multiple mRNA transcripts through alternative splicing; however, alternative splicing is specific to hr-HPV E6 (generating E6*) and E7 ORFs. This process has not been detected in lr-HPV E6 and E7 ORFs [51]. E6 also activates Akt and mTORC1 and inhibits keratinocytes differentiation [52]. Hr-HPV E7 induces cell transformation and immortalization, binding to tumor suppressor pRb family members p107 and p130 and inducing their degradation [53], and activates and promotes Akt signaling [54]. In contrast, the differences and similarities with lr-HPV are that lr-HPV E5 activates EGFR and is unknown if E5 induces PI3K. E5 does not downregulate MHC-class I. Lr-HPV E6 only binds to p53 but not degrades it, E6 does not activate hTERT, E6 does not has alternative splicing, and does not activate Akt signaling. Lr-HPV E7 has low affinity to pRb, p107, and p130, and does not induce their degradation. E7 does not induces cell transformation and immortalization. 

Although there are many differences between the lr-HPV and hr-HPV E5, E6, and E7 proteins as described above, these proteins converge in the dysregulation of cell cycle and signaling pathways for the maintenance of the viral genome and infection. This dysregulation is closely related to the autophagy process, as seen by the increase in autophagy in lr-HPV infection, and the decrease in autophagy in hr-HPV infection. Hr-HPV may repress autophagy as a strategy to enhance its ability to replicate, complete the viral life cycle, and increase infectivity.

## 4. Function of ncRNAs in the Regulation of Autophagy in Cervical Cancer Cells

A sizeable evidence base supports the correlation between molecular mechanisms involved in autophagy induction and those involved in tumor suppression, as well as between signaling pathways involved in autophagy inhibition and in oncogenesis. It will be important to characterize the role of tumor suppressor microRNAs and oncogenic microRNAs, as well as long non-coding (lncRNAs) and circular RNAs (circRNAs), in the modulation of autophagy. MicroRNAs can cause autophagy upregulation or downregulation by targeting genes or autophagy-related signaling pathways. Therefore, microRNAs have a large potential role in autophagy regulation and may serve as diagnostic and prognostic markers. MicroRNA involvement would suggest a broad involvement of autophagy in most human cancers. Current chemotherapeutic agents used for the treatment of cervical cancer have been shown to modulate autophagy by means of ncRNAs, and there have been significant recent efforts to design novel drugs and gene therapy approaches which directly or indirectly utilize ncRNAs to target autophagy and other cellular processes in cervical cancer [55] (Table 1). However, clinical treatment protocols employing these technologies are not yet available. 

Several studies have evaluated a treatment strategy using ribozyme (APOBEC) and antisense oligonucleotides to inhibit HPV E6 and E7 mRNA expression [70,71]. However, these approaches have low efficiency, short time of stability, and high cost of design and administration. An effective alternative strategy that can knockdown gene expression in a sequence-specific way and repress viral oncogenes at the posttranscriptional level in considerable magnitude, is with small non-coding RNAs [72]. The RNAi mechanism is a natural process by which gene expression in eukaryotic cells is silenced by microRNAs which cleave target mRNA. Previously, we reported that that selective silencing of HPV16 E6 and E7 oncogenes by siRNAs has significant biological effects on the survival of human cervical cancer cells [73]. We used HPV16+ SiHa cervical cancer cells transiently transfected with specific siRNA expression plasmids for HPV16 E6 and E7 oncogenes. In this model, we detected repression of E6 and E7 oncogene and oncoprotein expression and increases in p53 and hypophosphorylated pRb isoform protein expression, as well as features of autophagy and apoptosis morphology. These findings suggest that selective silencing of HPV16 E6 and E7 oncogenes by siRNAs has significant biological effects on the autophagy process, as well as on the survival of the cancer cells, and represents a potential gene therapy strategy in cervical cancer.

The tumor microenvironment plays a critical role in tumor cell proliferation and progression. As the tumor increases in size, it rapidly outgrows its blood supply, leaving tumor cells deprived of oxygen. Several studies suggest that hypoxia activates autophagy, and that hypoxia-inducible factor (HIF) is a key transcription factor that allows rapid adaptation to hypoxic stress. Thus, hypoxia has emerged as a pivotal factor in solid tumors associated with rapid growth, inadequate blood supply, tumor progression, and resistance to therapy. Hypoxia regulates the expression of a group of microRNAs in a cell type- and tissue-specific manner. Wan et al. identified hypoxia-induced miR-155 as a potent inducer of autophagy, which decelerates cell proliferation and blocks G1/S cell cycle progression in cervical cancer cells. Knocking down endogenous miR-155 inhibited hypoxia-induced autophagy [56]. They reported that miR-155 targets multiple molecules in the mTOR signaling pathway, including RHEB, Rictor, and RPS6KB2. Thus, they concluded that miR-155 is a key regulator of autophagy via dysregulation of the mTOR pathway. 

The mTOR signaling pathway senses and integrates a variety of environmental cues to regulate cellular homeostasis through mechanisms which include autophagy, and it is implicated in several pathologies including cancer [74]. Certain microRNAs suppress autophagic activity by targeting genes coding for ATGs under stress conditions, while other microRNAs promote autophagy by repressing important upstream signals in the autophagy pathway [56,75]. The miR-15a/107 cluster contains a series of microRNAs, including miR-15a, miR-15b, miR-16, miR-103, and miR-107 families. MiR-15a and miR-16 form two different clusters in mammals, and these two microRNAs are deleted or downregulated in many types of cancers [76]. Furthermore, miR-15a and miR-16 were shown to act as tumor suppressors which inhibit cancer cell proliferation by targeting various cyclins and CDKs, and induce apoptosis through downregulation of the antiapoptotic gene Bcl-2 [77]. Interestingly, miR-15a and miR-16 have also been reported to be potent inducers of autophagy. Rictor, a component of the mTORC2 complex, is directly targeted by miR-15a and miR-16. Overexpression of miR-15a and miR-16 or depletion of endogenous Rictor attenuates phosphorylation of mTORC1 and p70S6K and inhibits cell proliferation and G1/S cell cycle transition in human cervical carcinoma cells. Furthermore, it has been reported that miR-15a and miR-16 dramatically enhance the anticancer drug camptothecin (CPT)-induced autophagy and apoptotic cell death in HeLa cells [57].

Regarding anticancer drugs, hydroxycamptothecin (HCPT) is a derivative of camptothecin and represents a new generation of cytotoxic agents which target DNA topoisomerase I and have been used to treat various cancers with fewer side effects [72]. HCPT was found to induce autophagy by suppressing the expression of miR-30a in cervical carcinoma cells. Decreased expression of miR-30a is associated with HCPT-induced autophagy through modulation of the expression of LC3II and Beclin-1, proteins involved in autophagosome formation [78]. These findings highlight the importance of understanding the relationship between microRNA regulatory mechanisms and autophagy, which underly the cellular and molecular processes involved in the antiproliferative effects of anticancer drugs.

Pirarubicin (THP), another drug under study in the treatment of solid tumors, is an anthracycline with favorable antitumor efficacy and limited side effects. Wu et al. reported that THP induced a protective macroautophagy response in cervical cancer cells, and suppression of this type of autophagy enhanced the cytotoxicity of THP [59]. In this study, the authors found that upregulation of the autophagy-related gene ATG4B played an important role in THP-induced autophagy. THP triggered downregulation of miR-34c-5p, which was associated with the upregulation of ATG4B and autophagy induction; conversely, overexpression of miR-34c-5p decreased ATG4B levels and attenuated autophagy in THP-treated cervical cancer cells. These data suggest a miR-34c-5p-ATG4B-autophagy signaling axis, which may account for THP resistance among cervical cancer patients in clinical trials. Although these and other agents such as rapamycin, lithium, and chloroquine are clinically available and may be helpful for treating diseases associated with autophagy dysregulation, there are also genetic approaches to inhibit autophagy. These approaches include knockout of ATG genes by homologous recombination or knockdown by siRNA. Gene therapy studies have yielded additional information about the biologic roles of autophagy in cervical cancer.

Expression of the microRNA miR-21 has been found to be altered in almost all types of cancers and it is an attractive target for genetic and pharmacological modulation in various cancers. To identify the downstream cellular target genes of upstream miR-21, our group analyzed endogenous miR-21 expression in cervical intraepithelial neoplasia-derived cells. MiR-21 is overexpressed in these cells, and the tumor suppressor gene PTEN is one target of miR-21 [60]. Interestingly, when we analyzed the role of miR-21 in cell proliferation, we found that tumor cells in which miR-21 was silenced with siRNA exhibited a markedly reduced cell proliferation, as well as autophagy and apoptosis induction. These data suggest that PTEN mediates the biological processes affected by miR-21 in cervical cancer cells, including induction of cell death by autophagy and apoptosis.

Fang et al. performed a microRNA microarray analysis in cervical cancer tissues and identified a large number of microRNAs with differential expression in hr-HPV-infected tissues [61]. They reported that miR-224-3p is a candidate microRNA selectively upregulated in HPV-infected tissues and cell lines. Furthermore, they described how miR-224-3p regulates autophagy in cervical cancer tissues and cell lines. While overexpression of miR-224-3p inhibits autophagy in HPV-infected cells, knocking down endogenous miR-224-3p increases autophagy activity in the same cells. MiR-224-3p directly inhibits the expression of autophagy related genes such as the gene encoding the FAK family-interacting protein of 200 kDa (FIP200). These findings show that miR-224-3p regulates autophagy in hr-HPV-infected cervical cancer cells by targeting FIP200 expression.

Lu et al. investigated the functional role of miR-338 in autophagy and proliferation in cervical cancer [62]. They found that levels of miR-338 were decreased in cervical cancer tissues and cells, and inversely correlated with ATF2 protein levels, indicating that ATF2 is a direct target of miR-338. Furthermore, they demonstrated that restoring miR-338 expression inhibited proliferation in HeLa and SiHa cells. Interestingly, reduced expression of miR-338 directly led to increased autophagy in cervical cancer cells, which was similar to the mTOR signaling inhibitor rapamycin. They demonstrated that inhibition of miR-338 expression could decrease p-mTOR and p-p70S6 expression, suggesting that miR-338 decreases autophagy in cervical cancer cells by activating the mTor signaling pathway. The results indicate that miR-338 may inhibit cell proliferation and autophagy by targeting the mTOR signaling pathway via ATF2 in cervical cancer cells.

Both autophagy and microRNAs have been reported to participate in the process of ER stress, which triggers the unfolded protein response (UPR) to restore the normal function of the ER. However, the relationship between autophagy and microRNAs in the process of ER stress in cervical cancer cells is still not clear. Guo et al. described how miR-346, which was induced under ER stress, modulated autophagic flux in HeLa cells [63]. The authors reported that by regulating the process of autophagy, miR-346 reduced ROS levels in HeLa cells, thus protecting them from cell death following ER stress. Furthermore, they demonstrated that GSK3B was a target of miR-346 and participated in ER stress-related autophagy. Mechanistically, they demonstrated that miR-346 activated autophagy by interrupting the association between Bcl-2 and Beclin-1 in a GSK3B-dependent manner. They concluded that miR-346 plays a role in the induction of autophagy under ER stress in cervical cancer cells.

Tan et al. analyzed the regulation of miR-378 in the autophagy process and its role in metastasis during cervical cancer development [64]. They demonstrated that miR-378 expression was significantly upregulated in cervical cancer and cervical intraepithelial neoplasia III tissues when compared with normal cervical tissues. Increased expression of miR-378 promoted tumor migration and invasion in vitro and metastasis in vivo, while downregulation of miR-378 suppressed these effects in vitro. ATG12, an autophagy-related gene, was identified as a direct target of miR-378 whose expression is downregulated by miR-378. Furthermore, they found that in cervical cancer tissues with lymph node metastasis, miR-378 was upregulated and ATG12 was downregulated when compared with lymph node negative cases. These findings indicate that miR-378 functions as an oncogene by promoting metastasis in cervical cancer and promotes downregulation of ATG12, a gene involved directly in autophagy processes. 

Zhou et al. investigated the effects and downstream targets of miR-20a in the processes of proliferation, apoptosis, and autophagy in cervical cancer [65]. They reported that miR-20a was highly expressed in cervical cancer tissues and cells. Inhibition of miR-20a resulted in reduced proliferation, increased apoptosis, and downregulation of autophagic activity. Mir-20a targets the thrombospondin 2 (THBS2) gene, which codes for a matricellular glycoprotein involved in the regulation of multiple biological processes, which has been characterized alternately as a tumor suppressor or as an oncogenic factor. THBS2 expression was reduced in cervical cancer cells and tissues and was inversely associated with miR-20a expression. Knockdown of THBS2 eliminated the antiproliferative, proapoptotic, and anti-autophagic effects of inhibiting miR-20a in cervical cancer cells. These findings indicate that miR-20a promotes proliferation and autophagy and inhibits apoptosis by targeting THBS2 in cervical cancer cells. 

Zhao et al. investigated the relationship between miR-20a and cell proliferation and metastasis in cervical cancer [79]. They found that the expression level of miR-20a was significantly higher in cervical cancer patients than in normal controls, and that aberrant expression of miR-20a correlated with lymph node metastasis, histological grade, and tumor diameter. In anti-miR-20a cervical cancer cell lines established by lentivirus, inhibition of miR-20a expression prevented tumor progression by modulating the cell cycle, apoptosis, and metastasis in vitro and in vivo. Tissue inhibitors of metalloproteinase 2 (TIMP2) and ATG7, an autophagy-related gene, were shown to be direct targets of miR-20a. These results indicate that miR-20a modulates proliferation, migration, invasion, and autophagy in cervical cancer cells by targeting ATG7 and TIMP2. These data support the involvement of miR-20a in cervical carcinogenesis, especially lymph node metastasis, and in the autophagy process. 

Li et al. investigated the regulatory mechanism of miR-204 in proliferation, apoptosis, and autophagy in cervical cancer cells [66]. MiR-204 expression was significantly decreased in cervical cancer tissues and cell lines. Cell viability was suppressed while apoptosis was enhanced in C33A cells transfected with miR-204. In addition, the overexpression of miR-204 reduced the expression of Bcl-2 and LC3I/II proteins and increased the expression of Bax and Caspase-3 proteins in C33A cells. The authors reported that miR-204 regulates the expression of ATF2, an autophagy-related gene. When ATF2 was silenced in C33A cells, cell viability decreased, and the expression of ATF2 and LC3I/II were downregulated. Furthermore, the ectopic expression of ATF2 may attenuate miR-204-mediated inhibition of proliferation in C33A cells. In summary, miR-204 inhibits proliferation and autophagy and induces apoptosis in cervical cancer cells by targeting ATF2.

Cui et al. described the role of miR-106a in cell proliferation and autophagy and its putative target Lkb1 in cervical cancer [80]. They identified high expression of miR-106a in both HPV16-positive cervical squamous cell carcinoma tissues and cell lines, in association with malignant clinicopathologic parameters. The exogenous expression of miR-106a promoted cervical cancer cell proliferation while attenuating autophagy. Furthermore, Lkb1 was targeted by miR-106a and overexpression of Lkb1 neutralized the effect of miR-106a on proliferation and autophagy in cervical cancer cell lines. These findings suggest that miR-106a plays an oncogenic role in the AMPK–mTOR signaling pathway mediated by Lkb1 in the autophagy mechanism in cervical cancer cells.

Wang et al. analyzed the regulation of autophagy by miR-155-5p caused by hr-HPV infection in cervical cancer [81]. In hr-HPV positive cervical lesion tissues and cervical cancer cell lines, the expression of miR-155-5p expression was decreased. Overexpression of miR-155-5p promoted autophagy, whereas downregulation of miR-155-5p had the opposite effect. Furthermore, miR-155-5p downregulation suppressed LC3 and promoted P62 protein expression through promotion of the PDK1/mTOR pathway, whereas miR-155-5p overexpression recovered LC3 and suppressed P62 protein expression by suppressing the PDK1/mTOR signaling. These findings support the role of miR-155-5p in regulation of autophagy and clarify its effect on the PDK1/mTOR signaling pathway in hr-HPV infection in cervical cancer.

Zhang et al. investigated the effect of lr-HPV11 E6 on autophagy mediated by manipulation of the AKT/mTOR and Erk/mTOR signaling pathways [30]. The expression of HPV11 E6 in the cells activated the autophagy pathway. The increased autophagy activity was the result of decreased phosphorylation levels of mTOR, and decreased AKT and Erk phosphorylation. These findings represent a step toward understanding the pathogenic mechanism of lr-HPV infection in the development of genital warts.

Additional evidence supporting a role for autophagy in condyloma acuminatum comes from Wu et al., who described autophagy-related miRNAs and their targets in this condition [82]. They found that LC3 and P62 were increased in the local lesion tissue of condyloma acuminatum patients compared with healthy controls. Eighty-one differentially expressed miRNAs were identified, of which 56 were downregulated and 25 were upregulated. MiRNA-30a-5p and miRNA-514a-3p were associated with autophagy. Furthermore, the target genes of miRNA-30a-5p were identified as Atg5 and Atg12, and the target genes of miRNA-514a-3p were Atg3 and Atg12. These findings suggest that miRNA-30a-5p and miRNA-514a-3p expression affect the autophagy mechanism in the setting of lr-HPV infection, and play a role in the development of condyloma acuminatum.

Long noncoding RNAs (lncRNAs) are a class of RNA molecules with transcripts over 200 nucleotides in length which do not encode proteins. Instead, they regulate the expression of genes at various levels including epigenetic and transcriptional. lncRNAs have also been studied in the autophagy process in cervical cancer cells. Zou et al. evaluated the effect of paclitaxel on autophagy, proliferation, and apoptosis in human cervical cancer cell lines through the expression regulation of lncRNA RP11-381N20.2 [67]. By analyzing genome-wide expression profiles of chemotherapy-sensitive and insensitive patients with cervical cancer in the TCGA database, they determined that the expression of lncRNA RP11-381N20.2 was significantly lower in the chemotherapy-insensitive group, and RP11-381N20.2 expression positively correlated with total patient survival time. The expression of RP11-381N20.2 was decreased in cervical cancer tissues compared with normal tissues. The expression of RP11-381N20.2 negatively correlated with paclitaxel treatment time and dose and inhibited the proliferation of cervical cancer cells in a dose-dependent manner. Paclitaxel induced autophagy in a time- and dose-dependent manner in SiHa cells, and combined with RP11-381N20.2, significantly increased apoptosis in cervical cancer cells (Figure 1). These results allow to conclude that during paclitaxel-induced cell death in cervical cancer cells, the efficacy of autophagy may be affected by the expression level of lncRNA RP11-381N20.2.

Feng et al. determined the potential role of autophagy-related lncRNA in cervical cancer to construct an autophagy-related lncRNA signature for survival of cervical cancer [55]. The lncRNAs in CC were downloaded from The Cancer Genome Atlas (TCGA) database, and autophagy-related lncRNAs were identified through the co-expression of lncRNA genes and autophagy genes. Several autophagy-related lncRNAs with prognostic value (AC012306.2, AL109976.1, ATP2A1-AS1, ILF3-DT, Z83851.2, STARD7-AS1, AC099343.2, AC008771.1, DBH-AS1, and AC097468.3) were identified. Thus, this 10 autophagy-related lncRNA signature has prognostic potential for cervical cancer.

LncRNA HOTAIR is a trans-acting long non-coding RNA containing six exons in humans. In patients with cervical cancer, elevated HOTAIR levels are significantly associated with poor prognosis. Interestingly, HOTAIR plays an oncogenic role in cervical cancer by promoting cell proliferation, migration, invasion, and autophagy, inhibiting cell apoptosis, stimulating angiogenesis, accelerating cell cycle progression, and inducing epithelial–mesenchymal transition. Moreover, blockade of HOTAIR by artesunate or propofol shows promise for further development of this lncRNA as a potential therapeutic target in cervical cancer [83]. 

Shi et al. explored the role of lncRNA LINC00511 in the regulation of cell autophagy and apoptosis during cervical cancer development, through the transcription factor retinoic X receptor alpha (RXRA)-regulated expression of phospholipase D1 (PLD1) [68]. LncRNA LINC00511 and PLD1 expression were elevated in cervical cancer cells and tissues. LINC00511 was shown to bind RXRA, and overexpression of LINC00511 increased PLD1 mRNA and protein expression in cervical cancer cells. Introduction of siRNAs against LINC00511, RXRA, or PLD1 led to repression of proliferation and promotion of autophagy and apoptosis of cervical cancer cells. Furthermore, silencing of LINC00511 or PLD1 by siRNAs inhibited tumorigenesis in vivo in nude mice. These findings suggest that the lncRNA LINC00511 acts as an oncogenic lncRNA in cervical cancer via the promotion of PLD1 regulated by the transcription factor RXRA.

Circular RNAs (circRNAs) are a unique class of noncoding RNAs characterized by a covalently closed loop without any 5′-3′ polarity or a polyadenylated tail. Several studies have demonstrated that circRNA 0023404 plays a crucial role in the progression of cervical cancer. It has been reported to regulate miR-5047 and VEGFA in cervical cancer metastasis and chemoresistance [69]. CircRNA 0023404 knockdown attenuated cervical cancer cell invasion and lymphatic vessel formation of human dermal lymphatic endothelial cells (HDLEC). MiR-5047 inhibitor-transfected HeLa and SiHa cells enhanced invasion and lymphatic vessel formation of HDLEC cells. Interestingly, both circRNA 0023404 knockdown and miR-5047 mimic downregulated expression of VEGFA. Functional rescue experiments indicated that VEGFA acts as a key downstream factor in circRNA 0023404 and miR-5047-regulated invasion and lymphatic vessel formation. The autophagy-associated genes Beclin1 and p62 were dysregulated in circRNA 0023404-depleted or overexpressed HeLa cells. CircRNA 0023404 knockdown inhibited cell viability, which was abolished by autophagy inhibitor 3-MA in the presence of cisplatin. The rate of apoptosis was consistently elevated in circRNA 0023404-depleted cells and diminished in circRNA 0023404-overexpressed cells under treatment with cisplatin. These findings reveal a role of circRNA 0023404 in cervical cancer metastasis and chemoresistance through regulation of miR-5047 and VEGFA in the autophagy process.

## 5. Conclusions and Perspectives

In this review, we summarize the critical elements involved in the autophagy process and their interplay with HPV oncoproteins, as well as the potential mechanisms of posttranscriptional regulation of autophagy pathways mediated by ncRNAs during cervical cancer development. Whether autophagy functions in a pro-survival or pro-death manner may depend on the activity of tumor suppressors and/or oncogenic genes including HPV genes. There are several common upstream signaling pathways in autophagy that are related to HPV oncoprotein expression, including LC3II, LC3B, PIK3C, Beclin1, PTEN/Akt, SQSTM1/p62, UBC9, CD95, APO1, ATAD3A, PSAP, TPP1, STX12, TAP1, mTOR, RHEB, Rictor, and RPS6KB2, FIP200, THBS2, as well as ATG genes. Many of genes that code for these proteins are regulated by microRNAs, lncRNAs, or cirRNAs; such as miR-15a/107 cluster, miR-20a, miR-21, miR-30a, miR-34c-5p, miR-155, miR-204, miR-224-3p, miR-338, miR-346, miR-378, MIR-G-1, lncRNA RP11-381N20.2, lncRNA HOTAIR, lncRNA LINC00511, and circRNA 0023404. Importantly, a number of anticancer drugs have been shown to induce autophagy, and up- or down-regulation of certain ncRNAs may sensitize cervical cancer cells to treatment with fewer side effects for patients.

Hr-HPV oncoproteins inhibit the function of pro-autophagic proteins in two ways: direct interaction or promotion of degradation by the ubiquitin proteasome pathway. In recent years, a plethora of cellular proteins have been identified as targets of HPV oncoproteins. Limited interactions have been identified in the autophagy process. Thus, it is necessary to identify cellular proteins that are targeted by HPV E5, E6, or E7 oncoproteins during autophagy. An expanded understanding of HPV–host interaction will allow for the development of therapeutic strategies using microRNAs/lncRNAs/cirRNAs or anticancer drugs to inhibit HPV E5, E6, or E7 expression or block the interaction between cellular and HPV proteins and/or block the degradation of pro-autophagic proteins through the ubiquitin pathway. Furthermore, it is important to identify whether HPV E5, E6, or E7 oncoproteins upregulate or downregulate ncRNAs that play an important role in autophagy activation.

An unresolved question in targeting the autophagy process as a therapeutic strategy against cancer is which stage of the process to target: early steps in the autophagy pathway versus induction of autophagosomes versus the degradation of autophagosomes. The elements of autophagosomal pathways are involved as scaffolds in the induction of apoptosis and/or necrosis, and the accumulation of autophagosomes might alter these signaling pathways. Treatment strategies will differ by cancer type and whether autophagy stimulation is pro- or antitumorigenic for that cancer type and stage. Further studies are needed to delineate the most appropriate interventions to modulate autophagy in cervical cancer. 

A number of studies have described functional differences between the lr-HPV and hr-HPV proteins. Interestingly, among these differences are the effects of the proteins on the process of autophagy. In general, lr-HPVs induce autophagy during the development of genital warts, while hr-HPVs repress autophagy in the early stages of cancer development. In more advanced stages of cervical cancer, hr-HPV induce autophagy in response to the metabolic requirements of tumor cells. The repression of autophagy by hr-HPV may reflect a strategy that promotes favorable conditions for replicating the HPV viral genome in host cells. Treatments that induce autophagy could be aimed at preventing the development of cancer in lesions with hr-HPV infection. Conversely, therapeutic interventions to suppress autophagy could be helpful for eliminating warts in lr-HPV-associated lesions and in treating advanced stages of cancer associated with hr-HPV infection. The study of autophagy in cervical cancer represents a research challenge due to the complex interplay of the molecules that are involved in the autophagy process, and their relationship with the life cycle of HPV in both lr-HPV and hr-HPV infections.

Another interesting aspect in the study of autophagy in HPV infection in cervical cancer relates to the design of molecules or treatment strategies that are capable of inducing autophagy as a mechanism of cell death. It is important to consider whether the HPV viral infection is from lr-HPV or hr-HPV, and whether multiple HPV infections are present. In this context, it should be considered that persistent hr-HPV infection represents the main risk factor for the development of cervical cancer, which accounts for 90% of vaginal and anal cancers. In cervical cancer, the autophagy mechanism has a suppressive function to the carcinogenesis process. Cervical cancer patients with persistent hr-HPV16 and HPV18 infection have low levels of expression of Bcl-1 and LC3-II, which are relevant molecules in autophagy. This evidence suggests that suppression of autophagy could induce the process of cervical carcinogenesis. In addition, low expression of Bcl-1 and LC3-II, which are downregulated in part by miR-204, has been associated with a poor prognosis in cervical cancer patients compared to patients with CIN and healthy individuals, underpinning the role of autophagy as a tumor suppressor mechanism in this condition. Hr-HPV infection may induce errors in the autophagy induction mechanism and predispose to the subsequent development of cervical cancer.

During internalization of hr-HPV in the target cell, there is an interaction with EGFR, and this interaction induces the PI3K/Akt/mTOR signaling pathway, which negatively regulates the autophagy mechanism. Interestingly, miR-338 decreases autophagy levels in cervical cancer cells by activating the mTOR signaling pathway. In cervical cancer patients, there is an overexpression of EGFR. In summary, these findings suggest that the overexpression of EGFR may be associated with decreased expression of Bcl-1 and consequently induce a decrease in autophagy and the association with poor prognosis in cervical cancer.

By characterizing the mechanisms behind dysregulation of these critical factors and their impact on host cell autophagy, we advance understanding of the relationship between autophagy and progression from HPV infection to cervical cancer, including molecules and pathways which can be targeted in preventive and therapeutic strategies against cervical cancer.

## Figures and Tables

**Figure 1 cells-11-01323-f001:**
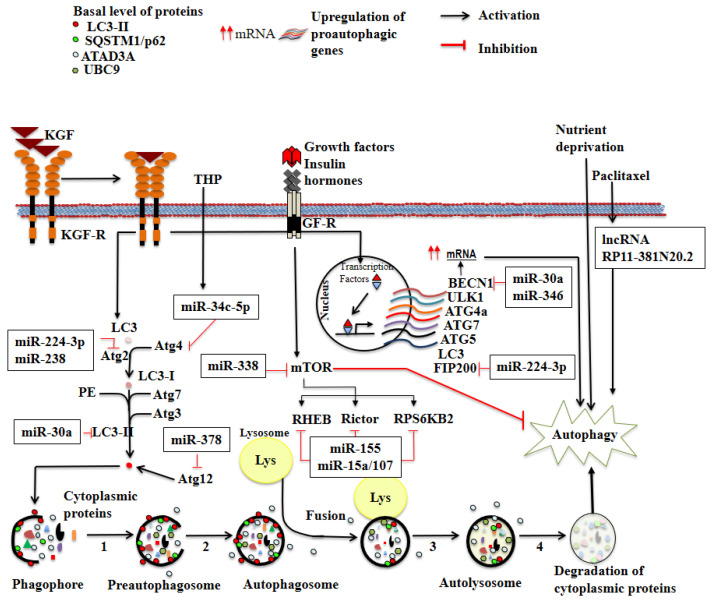
Regulation of the autophagic pathway by growth factors receptors. Autophagy is activated by nutrient deprivation, growth factor receptors (GF-R), infection, cancer, or other factors. Autophagy can be divided into several steps: (1) Cytoplasmic material is sequestered in a phagophore through the processes of nucleation and elongation of the isolation membrane induced by the autophagy-related protein LC3-II. (2) The edges of the phagophore then fuse to form the autophagosome. (3) The mature autophagosome membrane fuses with a lysosome to form the autolysosome, exposing the cytoplasmic content to lysosomal hydrolases. (4) Finally, proteins are degraded and recycled to the cytosol. The activation of some GF-R, such as the keratinocyte growth factor receptor (KGFR), induces autophagy through upregulation of several proautophagic genes or lipidation of LC3. LC3-II and p62 are important autophagy mediators, represented as red and green circles, respectively. LC3 is cleaved by ATG4 to generate LC3-I. LC3-I is conjugated to phosphatidylethanolamine (PE) by ATG7 and ATG3. The lipidated form of LC3-II attaches to the autophagosome membrane. The mammalian target of rapamycin (mTOR) kinase provides the major inhibitory signal to autophagy in the setting of nutrient abundance; conversely, miR-338 promotes autophagy by inhibiting mTOR. With the exception of miR-338, all microRNAs in the figure are involved in inhibition of autophagy. Pirarubicin (THP), an anticancer anthracycline, induces autophagy through downregulation of miR-34c-5p which upregulates the autophagy-related gene ATG4B. The chemotherapeutic agent paclitaxel interacts with the long noncoding RNA RP11-381N20.2 to induce autophagy. The role of additional microRNAs is indicated in each pathway. Red lines (
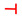
) indicate inhibition and black arrows (
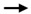
) indicate activation. Red arrows (
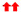
) represent increase in mRNA.

**Figure 2 cells-11-01323-f002:**
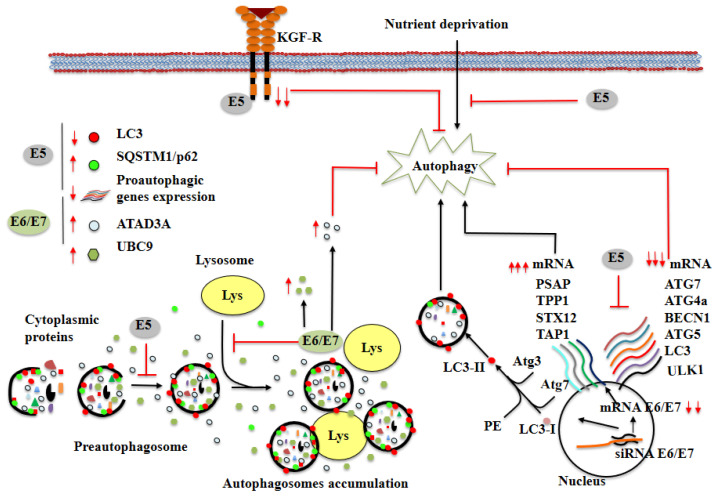
Modulation of autophagy by HPV16 E5, E6, and E7 oncoproteins. The image summarizes the effects of viral oncoproteins on both mRNA and cellular protein levels. HPV16 E5 oncoprotein inhibits autophagy activation by decreasing the levels of KGFR, downregulating the expression of autophagic genes, inhibiting the degradation of p62 and reducing the levels of LC3-II. E6/E7 expression increases the levels of UBC9 and ATAD3A proteins, protects cells from autophagy, and reduces fusion between the autophagosome and lysosomes. Silencing of E6 and E7 increases both the expression of autophagy-related genes and lipidation of LC3B, and reduces p62 protein levels. Red lines (
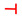
) indicate inhibition black arrows (
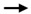
) indicate activation. Red arrows represent increase (
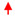
) or decrease (
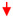
) in mRNA or cellular proteins, respectively. See the text for more details.

**Table 1 cells-11-01323-t001:** ncRNAs involved in the autophagy process in cervical cancer.

NcRNA	Target Genes	Biologic Effects	Reference
miR-155	RHEB, Rictor, RPS6KB2	Hypoxia-induced miR155 induces autophagy. Knocking down endogenous miR155 alleviates hypoxia-induced autophagy. The members of the mTOR pathway, RHEB, RICTOR, and RPS6KB2; are direct targets of miR155.	[56]
miR-15a/miR-16	Rictor	MiR-15a and miR-16 are potent inducers of autophagy. Rictor, a component of mTORC2 complex, is directly targeted by miR-15a/miR-16.	[57]
miR-30a	LC3II, Beclin-1	The decreased expression of miR-30a is involved in HCPT-induced autophagy in HeLa cells. MiR-30a directly target to Beclin-1.	[58]
miR-34c-5p	ATG4B	THP triggers downregulation of miR-34c-5p, associated with upregulation of ATG4B and autophagy induction. Overexpression of miR-34c-5p decreases the level of ATG4B and attenuates autophagy, accompanied by enhanced cell death and apoptosis in THP-treated cervical cancer cells.	[59]
miR-21	PTEN	There is an inverse correlation between miR-21 expression and PTEN mRNA level, as well as PTEN protein expression, in cervical cancer cells. Tumor cells exhibit reduced cell proliferation along with autophagy and apoptosis induction.	[60]
miR-224-3p	FIP200	MiR-224-3p regulates autophagy in cervical cancer tissues and cell lines. The overexpression of miR-224-3p inhibits autophagy in HPV-infected cells, while knocking down endogenous miR-224-3p increases autophagy activity. MiR-224-3p inhibits the expression of the FIP200 gene.	[61]
miR-338	p-mTOR, p-p70S6	Levels of miR-338 are decreased in cervical cancer tissues and cells, and negatively correlate with the protein level of ATF2. Inhibition of miR-338 expression decreases the expression of p-mTOR and p-p70S6, thus miR-338 decreases autophagy in cervical cancer cells by activating mTOR-signaling pathway.	[62]
miR-346	GSK3B	MiR-346 induced under ER stress modulates autophagic flux in HeLa cells. MiR-346 activates autophagy by interrupting the association between BCL2 and BECN1 in a GSK3B-dependent manner under ER stress.	[63]
miR-378	ATG12	ATG12 gene is a direct target of miR-378 and its expression is downregulated by miR-378 in cervical cancer cells. Thus miR-378 has a potential role in autophagy.	[64]
miR-20a	THBS2	The inhibition of miR-20a results in reduced proliferation, increased apoptosis and downregulated autophagic activity in cervical cancer cells. Thrombospondin 2 (THBS2) is a direct target of miR-20a.	[65]
miR-204	Bcl-2, LC3I-II, Bax, Caspase-3	The overexpression of miR-204 reduces protein expression of Bcl-2 and LC3I/II and increases protein expression of Bax and Caspase-3 in cervical cancer cells. MiR-204 regulates the expression of ATF2.	[66]
lncRNA RP11-381N20.2	Atg7, LC3A/B-II	The expression of RP11-381N20.2 is negatively correlated with the treatment time and dose of paclitaxel in cervical cancer cells. Paclitaxel combined with RP11-381N20.2 increases apoptosis of cervical cancer cell.	[67]
lncRNA LINC00511	RXRA	LINC00511 influences the occurrence of cervical cancer by upregulating PLD1 expression via recruiting transcription factor RXRA. SiRNA-LINC00511, siRNA-RXRA or siRNA-PLD1 trigger repression of proliferation and promotion of autophagy and apoptosis of cervical cancer cells.	[68]
circRNA 0023404	miR-5047, VEGFA	Hsa_circ_0023404 knockdown attenuates invasion of cervical cancer cells and lymphatic vessel formation of HDLEC cells. Hsa_circ_0023404 knockdown and miR-5047 downregulate the expression levels of VEGFA. Autophagy-associated genes (Beclin1 and p62) are dysregulated in hsa_circ_0023404 depleted and overexpressed in HeLa cells.	[69]

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
