# Peer review of "The Autophagy Process in Cervical Carcinogenesis: Role of Non-Coding-RNAs, Molecular Mechanisms, and Therapeutic Targets"

_cells, 2022, doi:10.3390/cells11081323_

Round 1

Reviewer 1 Report

The manuscript review the recent study on autophagy associated ncRNAs in HPV infection and cervical cancer. However,several  concerns need to be addressed in manuscript for publication.

1,the importance and current research progress in ncRNAs should be emphasized in introduction part, as the review title focuses on regulatory mechanism of ncRNAs in autophagy.

2, As we know that autophagy situation of different type HPVs infected cells  could be different,plsease check your cited references and clarify the difference.

3, patient with cervical cancer could be infected with multiple type of HPVs,so will these putative ncRNAs targets also applicable for most type of HPV infection?

Reviewer 2 Report

Lagunas-Martínez et al.'s review titled "The Autophagy Process in Cervical Carcinogenesis: Role of   Non-coding-RNAs, Molecular Mechanisms, and Therapeutic   Targets "has discussed the role of autophagy in cervical carcinogenesis and the function of non-coding RNAs (ncRNAs) in the regulation of autophagy. Furthermore, the effects of HPV E5, E6, and E7 oncoproteins on autophagy processes in cervical cancer development are also discussed.

This manuscript did not present a clear conceptual framework of the Discussion. Most of the part in the Discussion looks like a general description of other papers' findings but lacks their conclusion. I would like to see a clear structure on how autophagy affects cervical carcinogenesis via miRNA regulation and the therapy strategy accordingly. For example,

  1. The dual role of autophagy in cervical carcinogenesis;
  2. The consequences of induce or inhibitor of autopay in carcinogenesis;

More reference needs to be cited and discussed accordingly:

  1. Regulation of autophagy by high- and low-risk human papillomaviruses (PMID: 33590566).
  2. miR-106a Regulates Cell Proliferation and Autophagy by Targeting LKB1 in HPV-16-Associated Cervical Cancer (PMID: 32345599)
  3. MiR-155-5p inhibits PDK1 and promotes autophagy via the mTOR pathway in cervical cancer. ( PMID: 29627439).
  4. Human Papillomavirus 11 Early Protein E6 Activates Autophagy by Repressing AKT/mTOR and Erk/mTOR. ( PMID: 30971468)
  5. Dysregulation of autophagy-associated microRNAs in condyloma acuminatum.( PMID: 33905885).
  6. Autophagy in cervical cancer: an emerging therapeutic target. ( PMID: 23244072)
